# Early Bronchoscopy Improves Extubation Rates after Out-of-Hospital Cardiac Arrest: A Retrospective Cohort Analysis

**DOI:** 10.3390/jcm10143055

**Published:** 2021-07-09

**Authors:** Gregor S. Zimmermann, Jana Palm, Anna Lena Lahmann, Friedhelm Peltz, Rainer Okrojek, Florian Weis, Arne Müller, Tilman Ziegler, Alexander Steger, Bernhard Haller, Petra Hoppmann, Karl-Ludwig Laugwitz, Hubert Hautmann

**Affiliations:** 1Department of Internal Medicine I, School of Medicine & Klinikum rechts der Isar, Technical University of Munich, 81675 Munich, Germany; janapalm@me.com (J.P.); friedhelm.peltz@mri.tum.de (F.P.); rainer.okrojek@mri.tum.de (R.O.); florian.weis@mri.tum.de (F.W.); arne.mueller@tum.de (A.M.); tilman.ziegler@tum.de (T.Z.); alexander.steger@tum.de (A.S.); petra.hoppmann@mri.tum.de (P.H.); laugwitz@mytum.de (K.-L.L.); hautmann@tum.de (H.H.); 2Department of Cardiology, German Heart Center Munich, Technical University of Munich, 80636 Munich, Germany; AL@Lahmann-Kardiologie.de; 3Institute of Medical Informatics, Statistics and Epidemiology, Technical University of Munich, 81675 Munich, Germany; bernhard.haller@tum.de; 4Department of Internal Medicine, Klinik Ottobeuren, 87724 Ottobeuren, Germany

**Keywords:** out-of-hospital cardiac arrest, bronchoscopy, earl-onset pneumonia, ventilator associated pneumonia, post cardiac arrest syndrome

## Abstract

Background: Patients suffering from out-of-hospital cardiac arrest (OHCA) frequently receive a bronchoscopy after being admitted to the ICU. We investigated the optimal timing and the outcome in these patients. Methods: All patients who suffered from OHCA and were treated in our ICU from January 2013 to December 2018 were retrospectively analyzed. The data were collected from the patients’ medical files, and included duration of mechanical ventilation, antibiotics, microbiological test results and neurological outcome. The outcome was the effect of early bronchoscopy (≤48 h after administration) on the rate of intubated patients on day five and day seven. Results: From January 2013 to December 2018, 190 patients were admitted with OHCA. Bronchoscopy was performed in 111 patients out of the 164 patients who survived the first day. Late bronchoscopy >48 h was associated with higher rates of intubation on day five (OR 4.94; 95% CI 1.2–36.72, 86.7% vs. 55.0%, *p* = 0.036) and day seven (OR 4.96; 95% CI 1.38–24.69; 80.0% vs. 43.3%, *p* = 0.019). Conclusion: This study shows that patients who suffered from OHCA might have a better outcome if they receive a bronchoscopy early after hospital admission. Our data suggests an association of early bronchoscopy with a shorter intubation period.

## 1. Introduction:

Most patients with of out-of-hospital cardiac arrest (OHCA) and return of spontaneous circulation (ROSC) have a post-cardiac arrest syndrome (PCAS). The majority of patients with PCAS are comatose with impaired airway reflexes and mechanical ventilation is required after endotracheal intubation [1]. Despite ROSC many patients suffer from complications which are related to the underlying condition which lead to cardiac arrest and in addition to ischemia-reperfusion injury to the entire body and especially to the brain [2,3].

Due to the causes and consequences of resuscitation, pulmonary congestion in heart failure, thoracic compression with pulmonary contusion and volume therapy, impairment of the gas exchange and pulmonary infiltrations on chest X-rays are often found [1]. These pulmonary changes following OHCA can be misinterpreted as infiltrates on radiographic imaging [4,5]. In addition, patients with PCAS show an inflammatory activation, which in combination with pulmonary infiltrations on chest X-ray imaging may be interpreted as pneumonia [4,6,7]. 

However, pneumonia is commonly found during treatment in intensive care units with a subsequent impact on morbidity and mortality [4,7,8,9]. Within the first days early-onset pneumonia could be found in up to 65% of patient with OHCA [4,7,9]. There are multiple common mechanisms for this infectious complication. For example, in patients with post cardiac arrest syndrome loss of natural airway clearance, emergency intubation with possible aspiration, lung contusion, coma and mechanical ventilation are risk factors for pneumonia [5,6,10]. Moreover, therapeutic hypothermia can be considered as an independent risk factor for infection in patients with OHCA [2,4,9]. 

The impact on survival and neurological outcome seems to be minimal in early-onset pneumonia [7]. However, infections following OHCA are associated with prolonged ventilation time and intensive care unit stay [5]. Data on the influence of pulmonary infection and its therapy on the prognosis are inconsistent so far [9,11,12]. 

Prophylactic antibiotic therapy has shown no effect on the duration of mechanical ventilation, length of ICU stay, mortality or neurological outcome [7,9]. Data addressing the question of a default antibiotic regimen after cardiac arrest to prevent aspiration pneumonia is contradictory. It is discussed that the combination of infiltrative changes on radiographic imaging and increased inflammatory markers can also be misinterpreted as pneumonia [7].

After admission to the ICU for post-resuscitation care, patients oftentimes undergo bronchoscopy to improve the ventilation which is often impaired due to the consequences of resuscitation with severe cardiogenic shock, aspiration and airway management in the emergency setting as well as diagnostic endobronchial lavage for further microbiological analysis before starting an antibiotic treatment. However, the impact of a standardized bronchoscopic diagnostic on duration of ventilation, microbial surveillance and the intensive care outcome in the setting after OHCA remains unclear. There is no recommendation regarding bronchoscopy following OHCA or PCAS so far. Little is known about the importance of this diagnostic procedure as well as optimal timing and the effects on therapy regime and outcome of patients. Therefore, we aimed to characterize patients after OHCA presenting to a tertiary university center and investigate the influence of early post-resuscitation bronchoscopy (≤48 h) vs. late bronchoscopy (>48 h) with regard to the duration of mechanical ventilation and neurological outcome. 

## 2. Materials and Methods

### 2.1. Study Population and Endpoints 

This single-center, retrospective cohort study was conducted at the university hospital Klinikum rechts der Isar of the Technical University of Munich. We collected retrospective data of patients who were treated in our university tertiary medical center from 2013 to 2018. We included patients with ROSC after OHCA and transfer to our intensive care unit. Only patients with mechanical ventilation at administration and expected outcome >24 h were included in the analysis. The research protocol was approved by the local Clinical Institutional Review Board (Ethic Committee of the Technical University of Munich) and complies with the 1964 declaration of Helsinki and its later amendment. Data sets were anonymized after acquisition and before processing. Local ethics committee waived informed consent for this particular study. The primary outcome was the effect of early bronchoscopy (<48 h after administration) on duration of ventilation during hospitalization. Secondary outcomes were neurological outcome, assessment of bronchoscopic findings and microbiologic results. Neurological outcome was assessed by the Cerebral Performance Category (CPC) scale (CPC 1 good cerebral performance, CPC 2 moderate cerebral disability, CPC 3 severe cerebral disability, CPC 4 coma or vegetative state, CPC 5 brain death) on discharge or transfer to a special neurological rehabilitation clinic [13]. The timing for early bronchoscopy was determined to be within 48 h because the need for prolonged mechanical ventilation is usually evident within the first 48 h after OHCA. In addition, we included patients without emergency indication for bronchoscopy such as acute tube obstruction or malposition where immediate bronchoscopy would be necessary.

### 2.2. Data Collection

Demographic, clinical, treatment, and outcome data were reviewed following a standardized data collection protocol from the medical records. Patient records and data sets were reviewed retrospectively after anonymization and before processing. We evaluated hospital admission, duration of mechanical ventilation, time of bronchoscopy, cerebral performance status at discharge, antibiotic regimen and survival. If available, paramedical and early hospital information, predicting factors for long-term survival and former medical history in patients with out-of-hospital cardiac arrest were evaluated.

### 2.3. Bronchoscopy

The decision to perform early bronchoscopy in the ICU was left to the responsible physician’s discretion. If aspiration was documented during resuscitation, immediate bronchoscopy was performed. Flexible bronchoscopy was performed under sedation and mechanical ventilation via tracheal tube. The tracheal tube was initially placed by the emergency physician during resuscitation procedures after OHCA, except from a few cases where patients were intubated after admission. If the tracheal tube lumen was undersized a tracheal tube was inserted with a minimum size of 7.5 mm internal diameter. Bronchoscopy (BF Type PE2 or 1TH190, Olympus™, Japan) was carried out by an experienced physician following a standardized protocol [14]. A microbial specimen was obtained by mini-bronchoalveolar lavage (BAL). Large aspirates, segmental or lobar obstruction and blocking were documented. Secretion was removed by conventional suction.

### 2.4. Statistical Analysis

Statistical analyses were performed using the SPSS software package (version 26.0, SPSS Inc. Chicago, IL, USA) and R version 4.0.3 (© 2020 The R Foundation for Statistical Computing). All values are expressed as mean ± standard deviation (SD) or as median (inter-quartile range (IQR)), exceptions were indicated. Continuous variables were evaluated using the t-test and categorical variables using the chi-square test or Fisher’s exact test, as appropriate. A multivariate regression model was built, where potential confounders were chosen from a set of candidate confounders using a backward elimination algorithm. Statistical significance was assumed for two-sided *p*-values < 0.05.

## 3. Results

### 3.1. Patient Population

We included 190 patients after OHCA between 2013 and 2018 in this study. The mean age in our study population was 65.9 ± 15.2 years. There were 141 (74.2%) male patients and 53 (27.9%) patients had a non-shockable rhythm. In 22 patients (11.5%) there was an ongoing CPR on admission. Median time to ROSC was 15.0 (9.0, 22.5) minutes. On hospital admission 166 (87.4%) patients were intubated. Of 24 (12.6%) patients without intubation at admission, 5 had to be intubated in the shock room due to clinical instability. Most patients had myocardial ischemia as cause for OHCA, followed by rhythm event, which was defined as fast ventricular tachycardia or ventricular fibrillation without acute myocardial ischemia. All intubated patients received therapeutic hypothermia and prophylactic antibiotic therapy following standardized protocol. Baseline characteristics of the patients who survived the first 24 h are shown in Table 1. 

### 3.2. Outcome

After hospital admission, 26 (13.7%) patients died within the first 24 h in the shock room or intensive care unit. A total of 164 patients were included in the analysis. In 19 patients no invasive ventilation was necessary due to stable clinical condition. Four patients were on veno-arterial extra corporal membrane oxygenator and 14 patients had an intra-aortic balloon pump due to refractory cardiogenic shock. Pulmonary infiltrations were observed in 76 (46.3%) patients in chest X-rays. Atelectasis was described in six (3.7%) patients. After three days we observed 22 (13.4%) deaths and 111 (67.7%) patients were intubated and received invasive ventilation. Nine (5.5%) patients were extubated due to improved clinical and neurological condition. After five days 78 (47.6%) patients were still intubated and mechanically ventilated. After seven days 60 (36.3%) patients were intubated and received invasive ventilation. A total of 110 patients (66.7%) survived OHCA and could be discharged home or to a neurological rehabilitation program.

In 106 patients a CPC-score could be obtained. The CPC-score = 1 in 56 (52.8%) patients, CPC-score = 2 in 14 (13.2%) patients and CPC ≥ 3 was persistent in 36 (33.9%) patients. Patients with an CPC-score = 1 were significantly younger (60.0 ± 13.0 years vs. 67.7 ± 15.5 years, *p* = 0.027)

### 3.3. Bronchoscopy

Bronchoscopy was performed in 111 out of 164 patients who survived the first day. The study cohort is depicted in Figure 1. No patients had a complication regarding the bronchoscopic procedure. Of the 110 surviving patients, 79 patients underwent bronchoscopy. The median time of the bronchoscopy was 16.33 (IQR: 4.67–27.15) hours after hospital admission. 

In 11 patients an airway obstruction by secretion or coagulum was found. In 17 (15.3%) patients the endotracheal tube was repositioned due to unfavorable location close to the carina or in the right main bronchus. Microbial analysis of lavage revealed infection in 53 patients with Staphylococcus aureus (*n* = 23), Escherichia coli (*n* = 14) and Klebsiella pneumoniae (*n* = 13). The antibiotic therapy was changed in 28 patients after lavage. 

A significantly higher proportion of the surviving patients with late bronchoscopy after 48 h were intubated and ventilated at day five and day seven compared to patients with early bronchoscopy. As depicted in Figure 2, in 60 patients with early bronchoscopy within 48 h after hospital admission 27 patients (45%) were spontaneously breathing and 33 patients (55.0%) had mechanical ventilation. In 15 patients with bronchoscopy >48 h after hospital admission, 13 patients (86.7%) had mechanical ventilation and 2 patients (13.3%) were spontaneously breathing on day five. After 7 days 34 patients (56.6%) were spontaneously breathing and 26 patients (43.3%) were mechanically ventilated after early bronchoscopy, whereas 3 patients (20%) were spontaneously breathing and 12 (80%) patients were mechanically ventilated after late bronchoscopy >48 h. Early bronchoscopy was associated with a significant increase of extubated patients after OHCA after day five (OR 4.94; 95% CI 1.2–36.72, 86.7% vs. 55%, *p* = 0.036) and day seven OR 4.96; 95% CI 1.38–24.69; 80.0% vs. 43.3%, *p* = 0.019). These results were confirmed in the multivariate analysis. No difference could be found in PEEP or Horowitz index in patients with early or late bronchoscopy as depicted in Appendix A. In patients with late bronchoscopy, CRP on day five was significantly increased compared to patients with early bronchoscopy (198.5 ± 99.8 mg/L vs. 147.0 ± 91.8 mg/L, *p* = 0.034). 

Patients with late bronchoscopy had an increased CPC-score > 1 (86.7% vs. 54.4% OR: 0.20; CI: 0.02–0.82; *p* = 0.035). There was no significant effect on the combination of CPC 1 and 2 as favorable neurological outcome parameters after OHCA in patients with early bronchoscopy (CPC-score 1 and 2 vs. CPC > 2 (59.6% vs. 40.0%, OR: 0.45, CI: 0.14–1.44, *p* = 0.173). Information regarding CPC-score of the study population are depicted in Table 2. 

## 4. Discussion

The major finding of this study was that early bronchoscopy may have a beneficial effect in patients after OHCA in terms of duration of intubation and invasive ventilation.

Patients after OHCA have high mortality and even survivors require very complex and costly intensive medical care [15,16]. However, survival is limited and neurological impairment is also frequent [17]. So far there are only a few therapeutic approaches to significantly improve the outcome. In particular, therapeutic hypothermia has been able to improve outcome in this context. Nevertheless, many patients are still intubated and ventilated several days after OHCA and therapeutic hypothermia is a risk factor for respiratory complications [4,18]. Hence, we evaluated the effect of early bronchoscopy to obtain detailed information regarding airway and microbial surveillance in mechanical ventilated patients after OHCA.

In our study the population was similar to previous studies regarding age, sex and mortality [2,4,5,9,12,18,19,20]. The proportions of myocardial ischemia as the cause of resuscitation and shockable rhythm in our population were also similar to those found in the literature in larger studies [4,18]. The neurological outcome was in a range consistent with previous studies as well [4,18,21]. 

The mechanism for our finding of higher rates of ventilator-free patients after early bronchoscopy in patients with OHCA is unclear. One possible explanation is the reduction of pneumonia by preventing prolonged airway obstruction, e.g., by bronchial aspirations or secretions. Removal of bronchial aspirations may ameliorate ventilation to distal parts of the lung and therefore reduce substrate for ventilator-associated pneumonia. In patients with aspiration pneumonia early bronchoscopy could benefit the clinical outcomes of mechanically ventilated patients. Another advantage of bronchoscopic diagnostics is improved microbial sampling, which enables better control of antimicrobial therapy [22,23]. 

However, an accurate classification regarding pneumonia is difficult to obtain due to many confounding factors. Even in prospective studies, the diagnosis of pneumonia was subsequently changed in one quarter after adjudication [9]. Especially in patients with targeted temperature management it has been shown that pneumonia is difficult to determine and is not accurate in many cases [5,9,24]. In addition, our patients were on prophylactic antibiotics, which also makes it difficult to determine the presence of pneumonia [9]. The use of prophylactic antibiotics is not in accordance with the ILCOR recommendations, but due to the data regarding reduction of pneumonia rates, this is often established as standard practice at our and other institutions [3,9,11,12,25]. 

Accurate diagnosis and quantification of infectious complications in PCAS can be difficult to classify due to a variety of factors within a sepsis-like syndrome [6]. Thus, many laboratory and clinical signs of infection in the setting of PCAS and subsequent sepsis-like syndrome can only be evaluated in a complex manner. A number of mediators were shown to be elevated and these may mimic signs of systemic infection (e.g., IL-6, IL-10, IL-17, TNF-alpha, CRP, PCT and others) in patients with PCAS [6,16,26]. 

Clinical signs of pneumonia cannot be interpreted reliably under targeted temperature management. In addition, cardiac impairment and prolonged compressions following CPR with related contusion of the lung mimic infiltrative changes on chest X-rays and therefore an accurate quantification of pneumonia cannot be obtained in our population.

Another mechanism for the increased extubation rate in our study after early bronchoscopy is the early bronchoscopic assessment of endotracheal tube insertion. Malposition of the endotracheal tube close to main carina, in a main bronchus or intermediate bronchus is common in emergency medicine and can be easily corrected under direct visualization during bronchoscopy [27]. Malposition of the endotracheal tube is a risk factor for ventilator associated pneumonia and may aggravate ventilator-induced lung injury by volutrauma or barotrauma [27,28]. In our population, a correction of the tube was performed in 15.3% of patients, which is in the same range as previous studies with field intubation [27,28]. 

To date, there are few studies regarding bronchoscopy as a diagnostic or therapeutic procedure in the context of treatment following OHCA [10]. Previous studies have already discussed whether a beneficial effect can be found in cases of airway obstruction or aspiration [10]. Pneumonia is frequently found after PCAS, which can worsen the ventilatory situation. A risk factor for such early-onset pneumonia, in addition to aspiration, is therapeutic hypothermia [4,10,29,30]. Prolonged mechanical ventilation promotes incidence of late-onset pneumonia [24]. 

The main limitations of the study are the retrospective single-center study design and the small sample size. Hence, a retrospective study is only hypothesis generating. The study was conducted at a single university center, and local structures can also limit the interpretation of the results. However, our study is one of the largest evaluating bronchoscopic procedures after OHCA and the baseline characteristics found no difference between patients with early or late bronchoscopy.

Another limitation is that patients who were considered by the treating physician to have a severe limitation of prognosis may not have undergone bronchoscopy. This might be a possible bias of our study. To exclude these patients from the analysis, we evaluated only patients who survived more than 24 h. We determined the timing for early bronchoscopy within 48 h of hospital admission because the need for prolonged mechanical ventilation is usually evident within the first 48 h after OHCA, as pneumonia can often emerge during this time. However, due to difficulties in accurate determination of pneumonia in the setting of PCAS in previous studies we decided to evaluate the proportion of ventilated patients at day five and day seven as endpoint parameters. 

Our study is the first study investigating the topic of early bronchoscopy and its impact on duration of invasive ventilation. Confirmation of our retrospective study findings should be performed in larger populations with a prospective study protocol. However, routine bronchoscopy after OHCA and ROSC should be discussed in the clinical context, as prolonged hyperoxygenation, e.g., during bronchoscopy, may increase the risk of mortality and therefore the bronchoscopic procedure should be as rapid as possible [19]. In our population, we could not find any negative effect on mortality or cognitive impairment. Early bronchoscopy had no significant effect on the combination of CPC 1 and 2 as favorable neurological outcome parameters after OHCA. 

Early bronchoscopy within 48 h after hospital admission with OHCA was associated with higher rates of ventilator-free patients in our retrospective cohort study. Although the underlying mechanism of this association needs to be further elucidated, our finding emphasizes the importance of bronchoscopic diagnostics in the management of patients with PCAS. Thus, early bronchoscopy may improve weaning in patients with PCAS. 

## Figures and Tables

**Figure 1 jcm-10-03055-f001:**
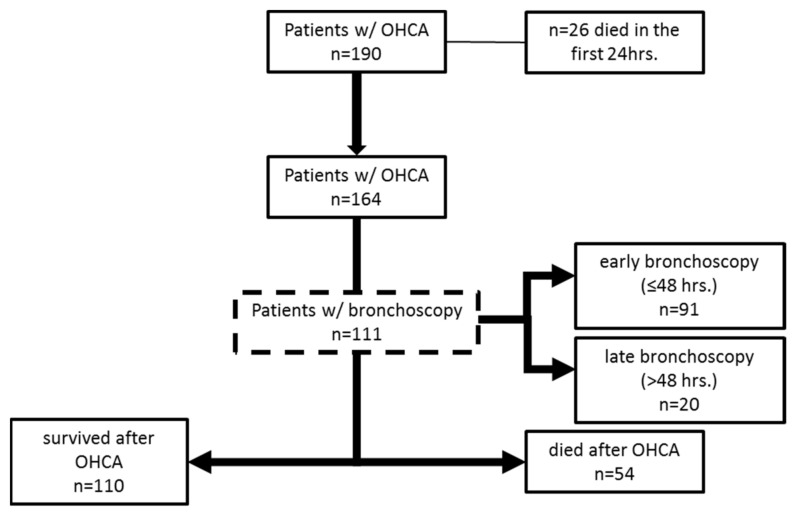
Patient flow diagram.

**Figure 2 jcm-10-03055-f002:**
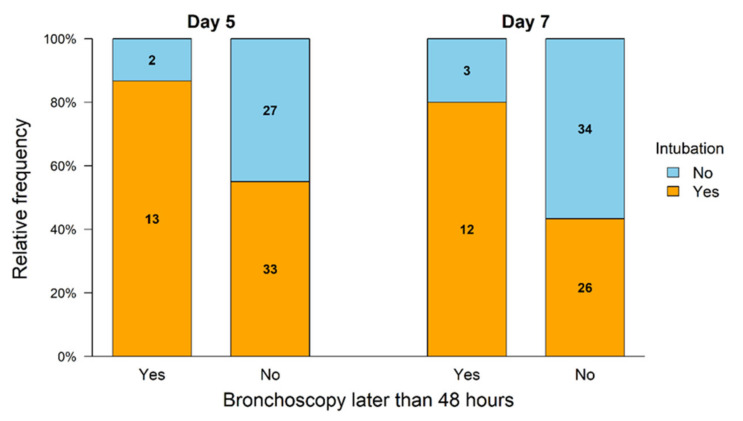
Intubation status of the surviving patients after OHCA. The group with early bronchoscopy <48 h showed a significantly higher proportion of extubation on day 5 (*p* = 0.036) and day 7 (*p* = 0.019) after OHCA compared to the group with later bronchoscopy.

**Table 1 jcm-10-03055-t001:** Baseline Clinical and Demographic Characteristics.

	All Patients *n* (%)	Survivors *n* (%)	Non-Survivor *n* (%)	*p*-Value
Patients	164	110	54	
Female	41 (25.0)	22 (20.0)	19 (35.2)	0.035
Age (years)	64.9 ± 15.0	62.1 ± 14.6	70.8 ± 14.2	<0.000
Ongoing CPR on admission	14 (8.5)	7 (6.4)	7 (13.0)	0.155
Shockable rhythm	107 (65.2)	81 (73.6)	36 (66.6)	0.001
Cause of OHCA				
Myocardial ischemia	91 (55.5)	66 (60.0)	25 (46.3)	0.097
Pulmonary embolism	7 (4.3)	3 (2.7)	4 (7.4)	0.163
Rhythm event	18 (11.0)	13 (11.8)	5 (9.3)	0.622
Unknown/other	48 (29.3)	28 (25.5)	20 (37.0)	0.162
Time to ROSC (min)	17.6 ± 15.8	15.0 ± 14.1	23.7 ± 17.8	0.012

**Table 2 jcm-10-03055-t002:** Neurological outcome of surviving patients at discharge. Data regarding CPC score was available in 106 (96.4%) of survivors.

CPC Score at Discharge	*n*	%	Age (Years)
CPC 1	56	52.8	60.0 ± 13.0
CPC 2	14	13.2	63.8 ± 16.4
CPC 3	28	26.4	67.7 ± 15.5
CPC 4	8	7.5	58.5 ± 15.9

## Data Availability

The data cannot be shared publicly because of the privacy of the individuals that participated in this study. Data are available from the Ethics Committee (contact via gregor.zimmermann@tum.de) for researchers who meet the criteria for access to confidential data.

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
