# Peer review of "Early Bronchoscopy Improves Extubation Rates after Out-of-Hospital Cardiac Arrest: A Retrospective Cohort Analysis"

_jcm, 2021, doi:10.3390/jcm10143055_

Round 1

Reviewer 1 Report

Thank you for the opportunity to evaluate your manuscript. For methodological reasons, I propose to consider responses to the following questions

  1. Whether endotracheal tube inflation pressure was assessed. Whether the reintubation procedure followed an established protocol, at what point the decision to reintubate was made and based on what parameters or symptoms.
  2. What do you mena Rhythm event in Tabe 1?
  3. Whether mechanical ventilation of patients was conducted according to an established schedule.
  4. Whether there were differences in mechanical ventilation, including ventilation parameters in the study and control groups
  5. What was the procedure of weaning from mechanical ventilation, was it done according to the established protocol
  6. Were comorbidities, age, and demographics similar in the study and control groups or were there any statistically significant differences.

The methodological issues of whether a comparison between study and control groups is possible, especially with retrospective data, should be reassessed. Retrospective analyses are subject to additional risk of error, so special attention should be paid to the issue of comparability of groups.

Reviewer 2 Report

The study by Zimmermann investigates the prognostic role of early versus late bronchoscopy in patients admitted to ICD after OHCA. Including 190 patients, the authors found early bronchoscopy associated with improved outcomes as compared to late bronchoscopy.

The manuscript is well-written and clear structured.

However, the following points are raised by this reviewer.

  • Since 26 patients were excluded from analyses after early death, those patients should not be presented in table 1.
  • Furthermore, baseline characteristic of patients treated with early versus late bronchoscopy should be presented. We need to know, if they differed in important baseline characteristics.
  • Do the authors have information on cardiovascular risk profile, important comorbidities (chronic kidney disease, coronary artery disease, heart failure, LVEF)? If yes, some data should be presented.
  • Furthermore, data on laboratory measurements should be given (initial lactate, pH, CRP). Please provide further data. Did this affect duration of ventilation?
  • Do the authors have information on catecholamine use? Kindly add.
  • Multivariable analysis should be performed to decrease the chance of possible selection bias, since patients with late bronchoscopy are sicker than those with early.

Round 2

Reviewer 2 Report

Congratulations - quality has signifficantly improved - no further comments.